# A Solution to Prevent and Minimize the Consequences of Accidents with Farm Tractors in the Context of Mountainous Regions with Low Population Density

**DOI:** 10.3390/s23187811

**Published:** 2023-09-11

**Authors:** Rui Alves, Paulo Matos

**Affiliations:** 1Departamento de Informática e Comunicações, Instituto Politécnico de Bragança, Campus de Santa Apolónia, 5300-253 Bragança, Portugal; 2Integrated Researcher at Centre in Digitalization and Intelligent Robotics (CeDRI), Polytechnical University of Bragança, Campus Santa Apolónia, 5300-253 Bragança, Portugal

**Keywords:** IoT, monitoring, prevention, farm tractor, low population density regions

## Abstract

Farm tractors have become a key part of daily routine agriculture, converting complex and time-consuming tasks into tasks that are easier to perform and less dependent on human labor, contributing directly to increasing the economic value generated by this activity sector, either by increasing the productivity or by making certain agricultural crops viable, which otherwise would not be sustainable. However, despite all the advantages, accidents with this type of equipment are common, often with critical and sometimes fatal consequences. The evolution of safety requirements of these machines has occurred at a good level; however, a significant part of the agricultural tractors in use are older models that do not have such solutions. Even in the new models, which contain such solutions, these are not always correctly used, and it is even common that they are turned off or simply not used at all. It is therefore natural that accidents continue to occur, a situation that is aggravated by other factors. Lack of situational awareness of the operators, which can result from advanced age, inadequate training, reduced sensitivity/respect for safety rules, or working on irregular terrain like mountainous areas, contribute to high-risk contexts that end in the loss of human life. The consequences of such accidents are clearly aggravated by the time it takes to assist the victims—either because accidents are simply not identified/reported immediately, or by the time it takes to locate and provide help to the victims. This is a scenario that is more common in mountainous regions and regions with low population density. The current paper, using NB-IoT, a set of sensors, and a web application, presents a conceptual toolset conceived to prevent accidents and minimize consequences (human and material) that can be applied to old and new farm tractors. The development was carried out taking the characterization of the farmers and the land in the region in which the authors’ research institution is located into account, which has the highest rate of fatal accidents with agricultural tractors in the country; it is a region of mountainous with a very low population density.

## 1. Introduction

The agricultural sector, with a continuous population growth, assumes a relevant role in society. Agriculture, in addition to being the main source of profit [1] in many regions, especially in regions of low population density, is also responsible for almost all raw material that is used to support the nutritional needs of the general population [2]. In the current context, maximizing the efficiency of this sector becomes critical. Recent research has shown that in the future, with a continuous demographic growth, the raw material coming from the agricultural sector may not be enough [3].

The introduction of farm tractors is a good example and an important step in the modernization process of this sector, and is being considered one of the greatest advances in the agricultural sector [4]. This type of machinery accelerates the execution of many tasks, increasing productivity and enabling the agricultural exploitation of new crops, thus promoting competitiveness and business opportunities.

Despite the advantages, the use of agricultural tractors has risks. The number of accidents with severe consequences is high, especially in mountainous and low-population-density regions [5]. The time it takes to assist the injured is crucial to avoid more critical consequences, like irrecoverable injuries or even death. In low-density population regions, farmers can spend long hours working alone in completely isolated places. In the event of serious accidents, reporting/identifying the occurrence of the accident is not always immediate. It is often as a result of the prolonged absence of the farmer that the suspicion arises that something may have happened—sometimes too late. The problem is aggravated by the time it takes to provide assistance to the injured, since the communications are not always the best, the rural roads are not always properly identified or have the best accessibility, the accidents often occur within private rural properties or in hard to reach places, and the advanced age and isolation of these communities may not help, among other peculiarities common to this type of region.

The reconstruction of accidents involving farm tractors is another problematic point. Most of the time, these accidents occur in remote areas without witnesses, making it even more difficult to clarify the circumstances in which the accident happened. Only analyzing the complexity of the lesions [6] provides important details about the main causes of farm tractor accidents, but it is not sufficient to determine the root cause. The ideal scenario to clarify these circumstances would be the provision of values for some indicators (e.g., speed, acceleration, slope, GPS position), something that, in current models of farm tractors, is not done yet.

Using a set of technologies and sensors, this work presents a solution that can help prevent accidents with agricultural tractors or, in more critical situations, provide information that may be fundamental to supply assistance to the victims and minimize consequences—ultimately saving lives. This information is equally important for reconstructing the circumstances of the accident, especially when the operator dies and there are no witnesses.

The rest of the paper is organized as follows: Section 2 details the context, including some statistics about accidents with farm tractors and information about the geographic region under study; the architecture and its working are detailed in Section 3; in Section 4 the preliminary results are detailed; and the conclusion and future work are described in Section 5.

### 1.1. Contribution

The core goal of the presented research is to increase the safety of farm tractors, in concrete terms:Help prevent accidents with farm tractors;Reduce the time of accident detection and assistance to the injured;Provide detailed information for the reconstruction of the accident, especially when the death of the driver occurs and there are no witnesses.

The idealized solution provides the following functionalities:**Danger Zones**: Setting danger zones and sending alerts to a predefined list of people is an important feature. Drivers often lose situational awareness and end up performing tasks in areas with an increased risk of accidents. The list will normally be made up of people from the circle close to the operator (emergency contact list), who want to be warned somehow of this type of situation. Alerting these people allows them to be aware of the situation, e.g., to pay attention to routines/habits that are not observed, or encourage them to contact the target (operator) to confirm that everything is fine and/or to alert them to the risk they are incurring;**Unusual Movements**: Based on client data, it is possible to detect unusual movements (e.g., rollover) made by the farm tractor. These movements can trigger alerts or be combined with other patterns for this purpose, such as whether the tractor remains immobile after an unusual movement;**Route Monitoring**: The main objective of this feature is to allow the reconstruction of the route made by the farm tractor to the position of a possible accident. It is an important aspect because most of the time, emergency teams do not know the regions, and spend time trying to find the exact location of the accident. This time might be important to save the life of the driver. This feature can also be used for general monitoring;**Operation Safety Area**: This is a feature that is not yet implemented, and is only included in the design of the architecture. The main objective is to analyze *the elevation parameter* in a certain area around the farm tractor during the execution of a certain task. A sound signal is emitted at the client in cases of unusual elevation levels;**Black Box**: This contains essential data (e.g., angular motion, speed, acceleration, GPS position, etc.) related to the tasks executed by the farm tractor. Although they are sent to the server, they are also stored on the monitoring device so that, in the event of an accident, they can help investigators determine the causes.

The research to conceive a solution considered a characterization of the population (age, social context, training, etc.), geography (land typology, accessibility, etc.), and agricultural tractors (age and safety equipment).

## 2. Motivation

The evolution of farm tractor design over time has been considerable [7]. Agricultural tractors are increasingly sophisticated equipment, containing advanced technological solutions, particularly with regard to comfort, safety, and efficiency. Some of these solutions are imposed by government authorities, such as encouraging the use of engines with lower greenhouse gas emissions, or the use of recyclable materials.

In terms of safety, designing a farm tractor has been a point of constant analysis, where points such as vibration, noise, and space for the operator to work have gained great relevance [8]. Additionally, points such as driving style, visibility, and the location of controls have also been considered in the design of these machines. However, their incorrect use has been at the origin of many of the accidents that occur with these machines. Among the main causes are [9]: inadequate design of the control cabin, lack of knowledge/training (ignoring safety procedures), and tiredness (performing tasks for a long period of time).

Most fatal accidents with farm tractors occur in mountainous areas [10] and, in part, there is no way to avoid farming in such regions. For example, in Italy [11], 76.8% of the territory is classified as mountainous. Another good example is Spain, where the region of Asturias [12,13], characterized by a relatively steep slope, has a high level of accidents with agricultural machinery.

The solution presented is focused essentially on regions with a low population density and a mountainous terrain. As is described in Section 2.1, the selected case study is the Trás-os-Montes region, in the north of Portugal, which is a region with these characteristics.

### 2.1. Study Case Region

Portugal is one of the three European countries [14] where farm tractor accidents cause the most deaths. The Trás-os-Montes region, in the north of Portugal, has the highest rate of accidents with farm tractors [15,16]. A reduction in these types of accidents involves the prevention and awareness of the dangers of driving these machines. However, as evidenced by data published in [14], these actions do not have a sufficient impact on eradicating or, at least, decreasing the occurrences of accidents. Analyzing the number of occurrences in the category of accidents in private property with agricultural vehicles, in 2021, the district of Bragança (part of Trás-os-Montes) occupies the first place, where 74% of these accidents occurred with farm or forestry tractors. The same source of information points out that 67% of victims are over 60 years old, and that the two main causes of accidents are uneven terrain and loss of control over the vehicle. This information is important to the scope of this work because, beyond mountains and low-density population characteristics, this region is also characterized by the older age of farmers, which is an important factor for the occurrence of accidents [17].

The old age of farm tractors [18] is another problem in this region. Despite the progress made in renewing the Portuguese fleet of agricultural tractors, a considerable number of tractors and agricultural machines are beyond the recommended service life (10 years), which has an impact on maintenance costs, but also increases the risk of accidents, whether because of the state of use of the equipment or because they are older models without the most recent protection innovations.

It is possible to establish a correlation between the age of farmers and the age of farmer tractors. Situational awareness may decrease with increasing age-related cognitive deficits and/or diseases [19] which, associated with the complexity of the task, irregular/mountainous terrain, and the use of older farm tractor models, often without the latest safety requirements, may expose the driver to high-risk contexts. A scenario often aggravated by the loss of situational awareness, overconfidence, or both, makes drivers accept the risk and perform tasks under such conditions.

Another problem is the time that is needed to identify a possible accident, where the early arrival of emergency services, in the majority of occurrences, is fundamental to preserve life or avoid more critical consequences. Normally, the mountainous and remote regions are associated with low population density, which delays the identification of accidents and severely compromises the chances of survival in case of serious injuries [20]. Moreover, accidents sometimes happen in hard-to-reach regions, where the rescue teams can find or access the location in useful time only with the help of people who know the region.

### 2.2. Related Solutions

Rollover Protective Structures (ROPS), shown in Figure 1, are one of the most well-known protective measures used in farm tractors to protect the driver in case of overturning, which, when used correctly, can practically avoid fatalities. Despite the level of safety that this type of mechanism offers, just in the USA, it is estimated that 1.6 million farm tractors do not have ROPS [21], where some of them were acquired without the mechanism, although they were designed to support it. Additionally, although ROPS performance standards are considered appropriate, it is not totally effective in eliminating rollovers. Regardless of the improvements in the design of this type of mechanism, where physical differences exist in the agricultural population have been considered (e.g., gender, height, strength, and joint mobility) [22], the number of accidents with farm tractors is still high. In addition, this type of mechanism does not prevent accidents but acts as a protection measure in the case of an accident, contributing to minimizing the consequences.

Moreover, the effectiveness of ROPS in accidents that occur in remote areas and low population density is questionable, especially if after the occurrence, the driver has serious injuries that require immediate medical care. The low population density and/or the fact that the area is remote can hinder and delay the detection of the accident, taking hours that are decisive to avoiding greater or irreversible damage, like the death of the victims.

In addition, despite the mandatory use of ROPS, regrettably, it is common for farmers to deactivate it, as it affects mobility and prevents certain operations from being carried out. As most agricultural tractors are used on private land, this situation is often beyond the control of the authorities. The solution proposed by the authors, due to the absence of data, automatically identifies any deactivation attempt.

Thus, and despite being undeniably useful, ROPS by itself is just part of what we envision as a solution, which should contribute to minimizing accidents but, if they do occur, minimize the consequences, providing the means to assist the injured in a timely manner.

No publications were found focused on the prevention or minimization of consequences with farm tractor accidents, and even less using IoT solutions, cloud, or edge computing. There is no way to make a comparison with the proposed solution. A large majority are studies about causes, consequences, and stats of farm accidents with machinery. Some that we consider most relevant are presented below:In [10], the authors present the results of research performed to evaluate the comprehension of safety signs depicting critical slopes, either in degrees or as percent values in a group of Italian agricultural machinery operators, while considering the possible influence of previous experience with agricultural machinery, previous incidents, and on-farm occupation;In [6], the authors present a study of the cause–effect relation between tractor overturns and traumatic lesions suffered by drivers and passengers, involving the analysis of the death scene, vehicle, traumatic lesions, and their comparison with the mechanical structures of the vehicle and the morphology of the terrain;In [24], the authors present the results of a study aiming to describe chores when farmers were either fatally or seriously injured and required emergency medical treatment as a result of overturns of tractors with or without rollover protective structures;In [25], the authors present a simulator to analyze accidents with farm tractors, testing scenarios that would not be possible to create in a real context.

## 3. Proposed Solution

Figure 2 represents the high-level architectural diagram of the proposed solution. In general, it is based on a client–server architecture, having a Broker, provided by AWS SQS [26], to interconnect the clients with the server. This approach allows bidirectional communication.

The server component consists of a database (MySQL) [27], a web server that allows access to the web management platform, and a set of background services for processing the data received from the various clients. The client, composed of a chip capable of communicating using the NB-IoT network, a small LCD screen, a memory card, and a set of sensors, must be physically installed on the farm tractor.

The use of the NB-IoT network is supported by the fact that it is a low-cost technology; has low energy consumption; and has long-range coverage, which makes it ideal for agricultural areas. It is part of the LTE network, and any place with network coverage is a good point for NB-IoT that significantly improves an IoT device’s energy consumption. In 2019, NB-IoT was already available in 69 countries [28,29]. As a mobile IoT network, security is not a problematic point, it uses a licensed spectrum and secure communication channels. Moreover, the mobile operators encrypt data and, in some cases, they use Virtual Private Networks (VPNs) and gateways with Access Point Names (APNs). Other important security points are Data over NAS (DoNAS), Non-IP Data Delivery (NIDD), or white-lists [30].

Moreover, it is important to note that the server component, the Broker, and some of the mechanisms used in the solution are provided by AWS Cloud. The use of the cloud plays an important role in reinforcing the availability and scalability and in future versions, and will facilitate the use of edge computing. To understand in more detail how each point described in Section 1.1 is supported, Section 3.1 and Section 3.2 will provide more technical details about the functionality of the client and server components, respectively.

### 3.1. Client Component

In Figure 3, it is possible to see the composition of the client in detail, which is composed of an Arduino MKR NB 1500 (Arduino manufactures, Monza, Italy) [31,32] (the only version in the Arduino family that supports NB-IoT communication), a GPS sensor (Rua de Cabanas No 600, Gondomar, Portugal) [33], an MPU6050 sensor (ElectronicWings, Pune, India) [34], and an external memory card (Rua de Cabanas No 600, Gondomar, Portugal) [35]. In general, the GPS Sensor is responsible for providing the GPS coordinates in each measurement; the MPU6050, which consists of a three-axis accelerometer and three-axis gyroscope, provides data related to acceleration and angular motion, used for prevention, but also to identify potential accidents (overturns, crash/collision, and the like); and the memory card is used to save data locally for a period of 24 h, ensuring that, in the occurrence of an accident, data can be obtained locally (safeguarding against a situation where the data has not yet been sent to the cloud, which can occur if there is no connectivity to the cloud). Moreover, it can be useful for investigators to identify the reason that cause of the accident and to reproduce the route made by the tractor from the moment it left the farm to the place where the accident occurred.

The client registration process is made by connecting the Arduino to a small LCD and a keyboard (not represented in Figure 3). This process consists of putting three parameters on the chip: the license plate of the farm tractor, the security code (which must be confidential), and the frequency to send data to the cloud.

In Figure 4, it is possible to analyze the flowchart of actions to register a client. After setting the license plate of the farm tractor, the security code, and the frequency to send data to the cloud, the request is submitted for approval on the server. In addition to the data defined by the user, the *UUID* of the chip that will identify the farm tractor is added to the request. After sending the request, the client will be waiting for the confirmation of registration by the server.

In Listing 1, it is possible to analyze the JSON response sent by the server after the confirmation of a registration request. The first three fields represented in this listing correspond to the retransmission of the *license_plate*, *uuid*, and *security_code* values sent at the time of submission of the registration request by the server. Sending this information to the client is an additional security level, ensuring that the client only starts sending the data after an explicit confirmation from the server. This confirmation implies the validation of the previous values, as well as an additional boolean field (*is_approved*) that effectively guarantees the approval of the application.

**Listing 1.** JSON received by client after server confirms the registration.{    "security_code" : "00XX",    "uuid" : "XYZT",    "license_plate" :  "00-XX-00",    "is_approved" : true,}

Data processing has two stages. In Listing 2, all fields that are used in both processes are illustrated. This information is generated by the client every *t_s_* (which usually corresponds to 1 s, but this can be configurable). The latitude and longitude can be found in the *latlon* parameter, corresponding to the value of these two measures at each sample. The *timestamp* corresponds to the moment in which the data was recorded by the client (Unix time). Additionally, the *acc_values* parameter represents the value of the three axes(x,y,z) for the acceleration. Finally, the *ang_values* parameter represents the value of the three axes(x,y,z) for angular motion.

**Listing 2.** JSON generated by client(s).{    "latlon" : [73.9012, −8.91347],    "timestamp" : 1686398073,    "ang_values" : [1, 0.5, 1],    "ang_high" : false,    "acc_values" : [1, 0.5, 1],    "acc_high" : false }

The first stage starts, before registering the record in the client’s local storage, when there is a verification of the values of the parameters *acc_values* and *ang_values*. For both parameters, the verification is made by comparison between two consecutive samplings. If the difference obtained in angular motion (*ang_values*) is greater than *x* rad/s, the *ang_high* parameter passes to *true*. On the other hand, in the case of the parameter *acc_values*, it is the parameter *acc_high* that passes to value *true* if the difference is greater than *y* m/s^2^. In both situations a beep is emitted, through the buzzer represented in Figure 3, with different patterns, to alert the driver to each of the situations.

The second stage is when the data is sent to the server. In Listing 3, the structure that is sent to the server by the client through the Broker is represented. The meaning of the *uuid*, *license_plate*, *security_plate* fields has been explained previously. They have been added to the JSON string as an additional security mechanism, as happened in the registration process. The sending period of this structure, as already explained, is defined at the time of the client registration. The *id_sent* field corresponds to the identifier of the data batch. As the data remains locally only for 24 h, its value corresponds to the Unix time of the moment the structure is added to the local storage. The Retry mechanism is supported by the *is_sent* and *is_confirmed* fields. The first field illustrates the number of tries the data batch has already been published in the Broker, and the second field describes whether the client has already obtained a confirmation of reception of that batch by the server. In this way, there is a mechanism that allows the synchronization of data in the case of sending failures. Finally, the *data* field corresponds to the values of the measurements made in the time range for a specific patch. For example, if the sending rate is set to 2 min and, as has already been mentioned, there is a measurement of values for local storage every 10 s, the *data* field will have 12 elements with the information represented in Listing 2.

**Listing 3.** Structure of the information, in JSON, sent to the server with each sending.{    "license_plate" : "00-XX-00",    "security_code" : "00XX",    "uuid" : "XYZT",    "id_sent" : 1686398073,    "is_sent" : 1,    "is_confirmed" : false,    "data" : [...]}

### 3.2. Server Component

A detailed view of the server can be analyzed in Figure 5. As mentioned, the server is composed of a database, a set of services in the background, and a web platform; however, the AWS SNS [36] is part of this component for sending SMS and/or emails.

Registration requests from clients are processed by *Register Services*. Data processing that supports the monitoring of farm tractor movements and all other functionalities is performed by *Data Services*. As with the client’s local storage, the server also maintains data history of the last 24 h for each client.

The architecture of the *Register Services* and *Data Services* is shown in Figure 6. These services were implemented using a microservice approach, using SpringBoot [37], and their execution is supported by AWS EKS [38]. This service allows, whenever the conditions of execution and system load require, the creation of new instances, something that can be performed automatically and/or based on rules. Based on type and/or client, the Gateway API is responsible for routing requests to the correct microservice instance. Moreover, on the *Data Services* side, an instance is created by each client, where all actions related to the processing of data will be carried out, via the Broker, for that client.

The data processing begins with the registration of the farm tractor on the web platform. As required fields, the user must provide the farm tractor license plate, the security code, and the location of the farm (the farmer’s work base as GPS coordinates). There is a set of optional data, such as the name and age of the usual driver, among others that could be supplied.

The client confirmation process on the server is simple. Each client publishes their registration data to the Broker (see Section 3.1). The information is read and processed by the *Register Services*. Then, a search is made in the database, trying to find the record equivalent to the information provided by the client, i.e., checking that the license plate and the security code received from the client are already registered in the database. It is important to note that the license plate and security code provided when creating the registry on the server must be exactly equal to the values provided in the client registration process. Finally, the server informs the respective client, via the Broker, whether or not the request was approved.

In Figure 7, it is possible to verify, in more detail, the set of actions that *Data Services* perform for each data batch received from the clients. In the first stage, the asynchronous waiting for new data occurs. When a data batch is published to the Broker, the client is identified and validated by the license plate and security code that comes in the received data. If this validation fails, the data is automatically discarded. In the case of a valid confirmation, a database search is performed to obtain the list of defined danger zones for the identified client, analyzing whether the received GPS coordinates are within any of these zones. When it has identified that the farm tractor is performing tasks in one of the defined danger zones, asynchronously, the list of emergency contacts is loaded, and, using AWS SNS, a text message is sent and/or email to each user in that list. The process continues with the analysis of the movements of the farm tractor where the acceleration and angular motion values are verified for all axes (x,y,z). If these values exceed the defined threshold, or if there is excessive variation in consecutive samplings, the movements are considered unusual, also triggering asynchronously, as in the previous step, and sending an alert message to the emergency contact list. Finally, the data is stored in the database, and the process goes back to the beginning.

The web platform, built using ReactJS [39], has a set of well-defined features: registration of the farm tractor; management of danger zones, including definition, check, and delete operations; management of the emergency contact list, including definition, check, and delete operations; and access of the route of the last 24 h made by the farm tractor.

The definition of danger zones is performed using the Drawing Tools API of Google Maps [40]. As can be seen in Figure 8, the danger zones are identified by a black rectangle, where the latitude and longitude of the points of the defined area are obtained, which will be saved in a database and used in the algorithm explained above.

Figure 9 represents another feature supported by the web platform: the possibility to consult and share the last route (A: Initial Point, B: Last Point Known) of the farm tractor since it left the premises, being always limited to the 24 h time range.

### 3.3. Other Features

Although not yet implemented, the solution will soon allow analyzing, in real-time, the movements of the farm tractor and emitting alerts (e.g., a beep for the driver) whenever is detected that the farmer is performing tasks on areas with high slopes, as shown in Figure 10.

This feature will make use of the Google Elevation API, which provides the degree of slope of the terrain based on geographic coordinates.

## 4. Results

The evaluation of features such as the detection of movements that can be considered an accident was not performed because we were not able to recreate such conditions. Setting a value is easy, but defining the right value, which implies its validation, is very complex. The rollover depends on many variables, such as the angular velocity and weight distribution of the tractor, the slope of the land, and the load it carries, among other variables. Being fundamental for the correct functioning of the solution, it does not, however, prevent validation of the architecture and operational workflow.

Figure 11 represents a chart containing the latency values in the first phase of the project development. The test was conducted in a region with good NB-IoT coverage, and the server has been put in the Paris region on AWS because it is the region closest to the geographical area under study. The values shown in Figure 11 used the data structures presented in a Listing 3. The latency values obtained have shown that latency may not be a problem.

Validating danger zones and sending alerts to the emergency contact list has already been performed. Figure 12 represents the text message received by a user from the contact list of the farm tractor with license plate *XX-00-XX*, identifying that the driver is performing operations in a defined danger zone (*Zone 1*), and alerting the person who received the message to take the necessary safety actions. In addition, the analysis, using a web platform, of the last route performed by the farm tractor was also validated (see Figure 9).

## 5. Conclusions

In recent years, accidents with farm tractors have been a constant concern of government authorities. The introduction of new safety procedures has been made with some frequency; however, despite all efforts, the number of accidents with these machines continues to be considerable, raising the need for better solutions to mitigate accidents with these machines.

The most critical points of analysis in this work have gone through are the execution of tasks in isolated and remote regions, where, when the accident occurs, it is often only identified after a long time, considerably increasing the probability of death in the event of serious injury; and the overestimation of the capacities of the operators, leading them to carry out tasks in dangerous contexts, increasing the risk of accidents.

The psychosocial capacities of the drivers of these machines play an important role [41], having a considerable impact on the success and/or lack of success of the preventive procedures that have been implemented over time. In some countries, such as Portugal or the USA [42], the procedures adopted have been based on training. However, it is known that the ability to identify danger and situational awareness during the handling of a farm tractor varies from individual to individual. The use of ROPS is another important adopted action; however, its effectiveness depends on what could happen in the event of an accident (the time it takes to provide support to the injured person). It is evident that the need for other tools, such as the one presented here, which, in a continuous and agnostic way, gives alerts about potentially dangerous situations, signals potential occurrences of accidents as soon as possible and, in the event of an accident, provides valuable information for timely and effective assistance, minimizing consequences and saving lives.

The usability, the ability to detect precise changes in the movement of the farm tractor, and the detection of task executions in areas considered critical are the main contributions of the presented solution. An important detail is that the definition of these danger zones is necessarily made by people who know the terrain, know the characteristics of the farm tractor, and, even more importantly, know the profile of the driver. All this is critical to define, with precision, the areas where the farm tractor driver should not perform tasks.

As detailed in Section 4, in the current state of development, the exchange of messages between the various components has already been validated. A remote control tractor was used to try to evaluate some more critical situations; however, the conclusions of these tests are not valid because a lot of differences between this scenario and a real operation scenario exist. In addition, the web application is already built, making it possible to demonstrate all the features described in previous sections. Ending with the testing phase, it was demonstrated that the use of the NB-IoT network can be a problem in regions where there is no coverage. Despite the existing relay mechanisms in the solution, the delay in sending the data can be a critical problem depending on the scale; the data that is sent to the server to process is the basis of operation to identify and forecast possible uncommon movements and alert emergency contacts. In the current phase, there is no clear and effective solution to this point; however, the use of SMS can be considered, but may not solve the problem, due to the delay that happens between sending and receiving the message.

### Opportunities and Open Issues

Several opportunities for improvement and problems are still open in the solution presented. The detection of unusual movements in the current version, as explained, is performed using only a threshold. The use of deep learning models is not excluded. However, as is possible to see in Section 4, more studies are needed to understand the impact of latency on data processing and the best strategy to reduce it, because most of the time, using deep learning approaches needs a high computing power.

The solution that could solve this issue, as well as others that are somehow limiting the current solution, is the use of edge computing solutions like rOpenCL [43]. The idea is to keep the whole part of model creation (learning) in the cloud, but moving the part of model use (forecast) to local units. This technical solution will be able to help provide a timely response, but also ensure part of the solution’s operation, even when the communication latency is high or simply when there is no access to the data network.

The points identified for future work, which need to be implemented and validated as soon as possible, are:Using machine learning algorithms to identify rollover situations;Creating a mobile application to receive push notifications;Validating the system in more realistic scenarios;Implementing slope degree monitoring functionality in an area around the farm tractor;Enabling the emergency contact to request more farm tractor information through the mobile application;Understanding the impact of latency on the detection of unusual movements and the forecast of possible accidents;Understanding whether SMS can be used in scenarios where NB-IoT coverage is weak or non-existent.

## Figures and Tables

**Figure 1 sensors-23-07811-f001:**
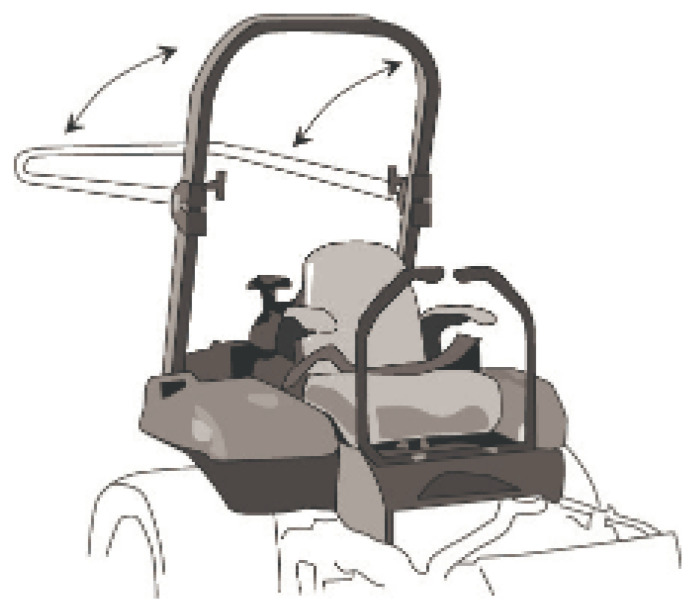
ROPS example (Reprinted/adapted with permission from [23]).

**Figure 2 sensors-23-07811-f002:**
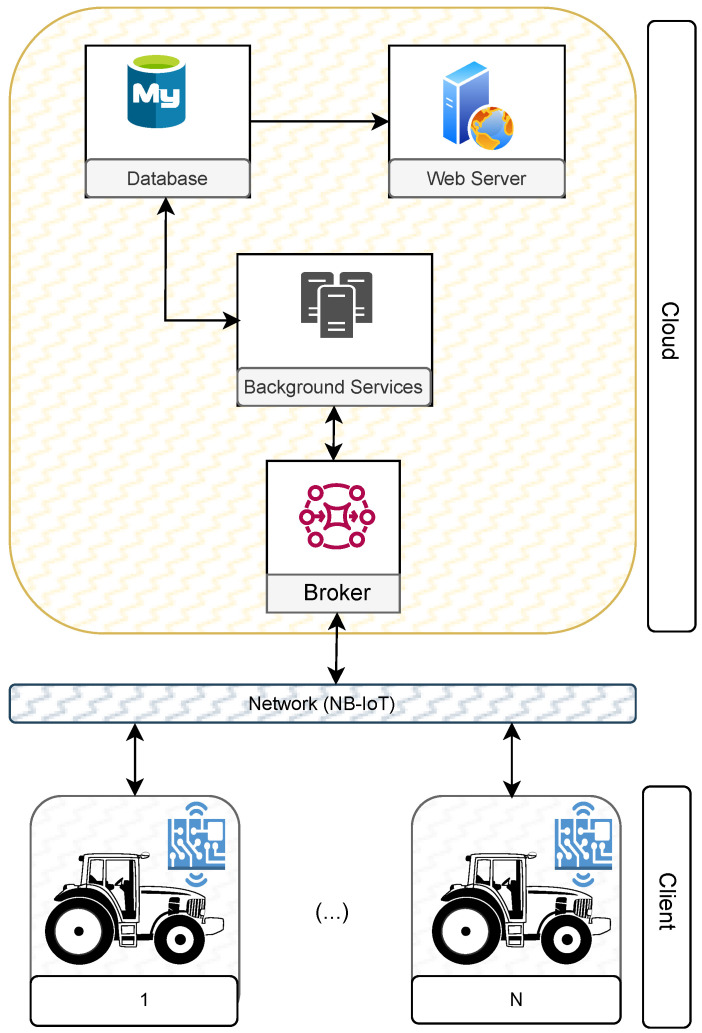
Solution overview.

**Figure 3 sensors-23-07811-f003:**
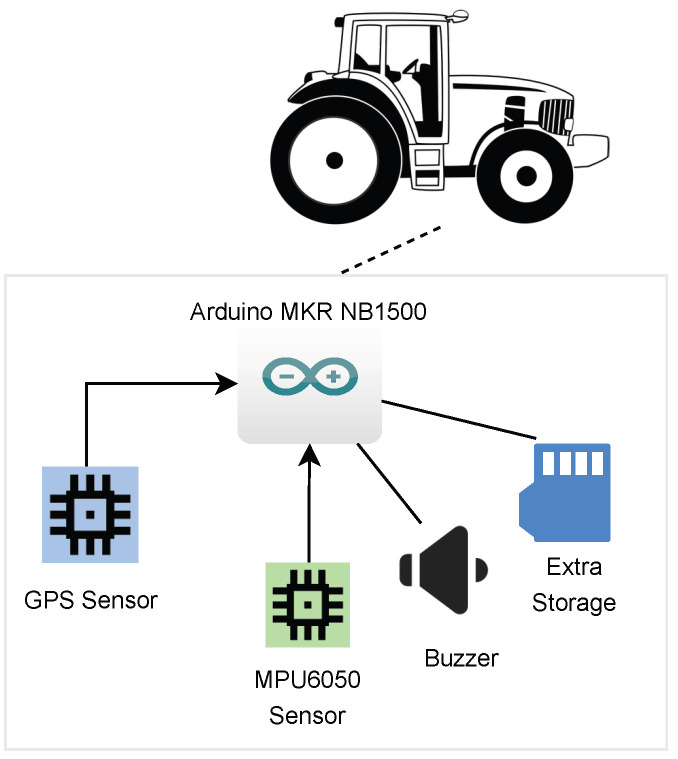
Client overview.

**Figure 4 sensors-23-07811-f004:**
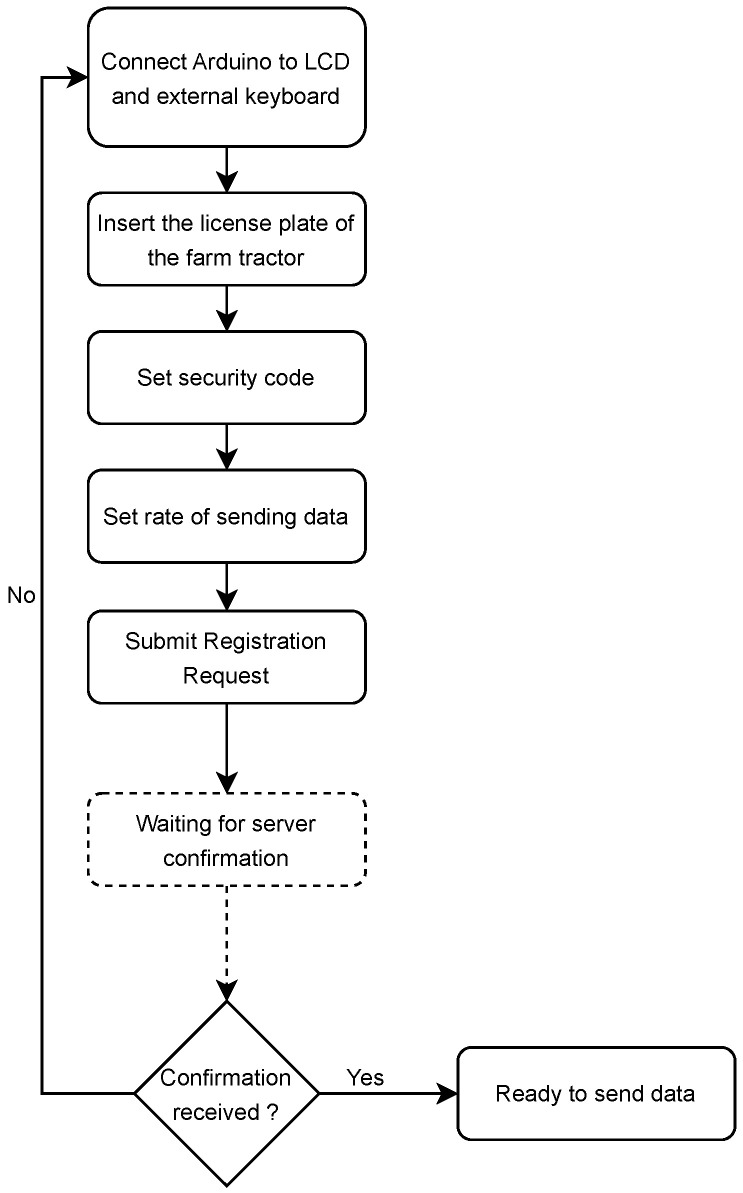
Flow chart of client registration process.

**Figure 5 sensors-23-07811-f005:**
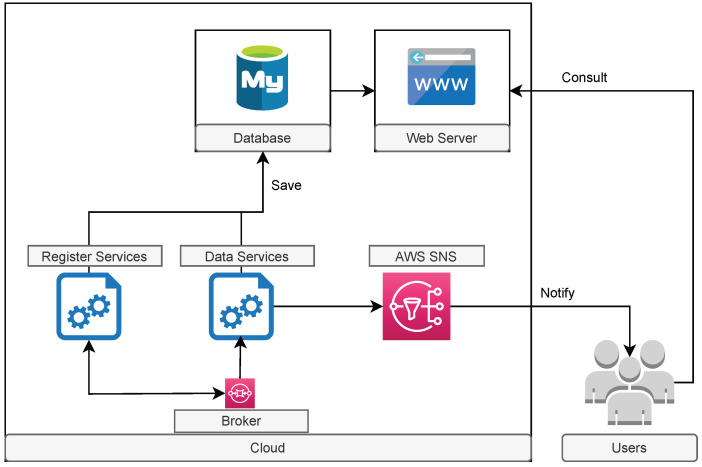
Server overview.

**Figure 6 sensors-23-07811-f006:**
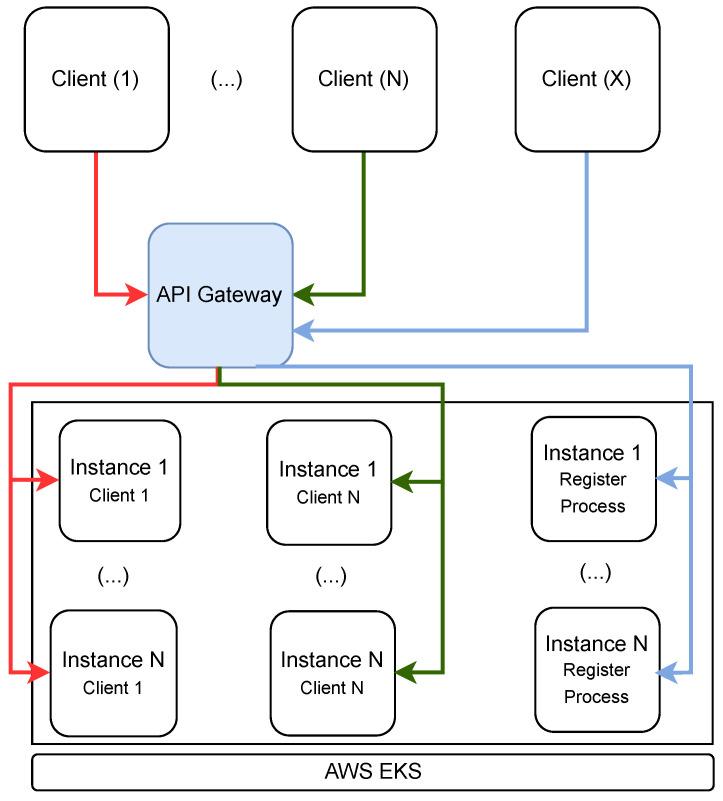
The architecture of the microservices.

**Figure 7 sensors-23-07811-f007:**
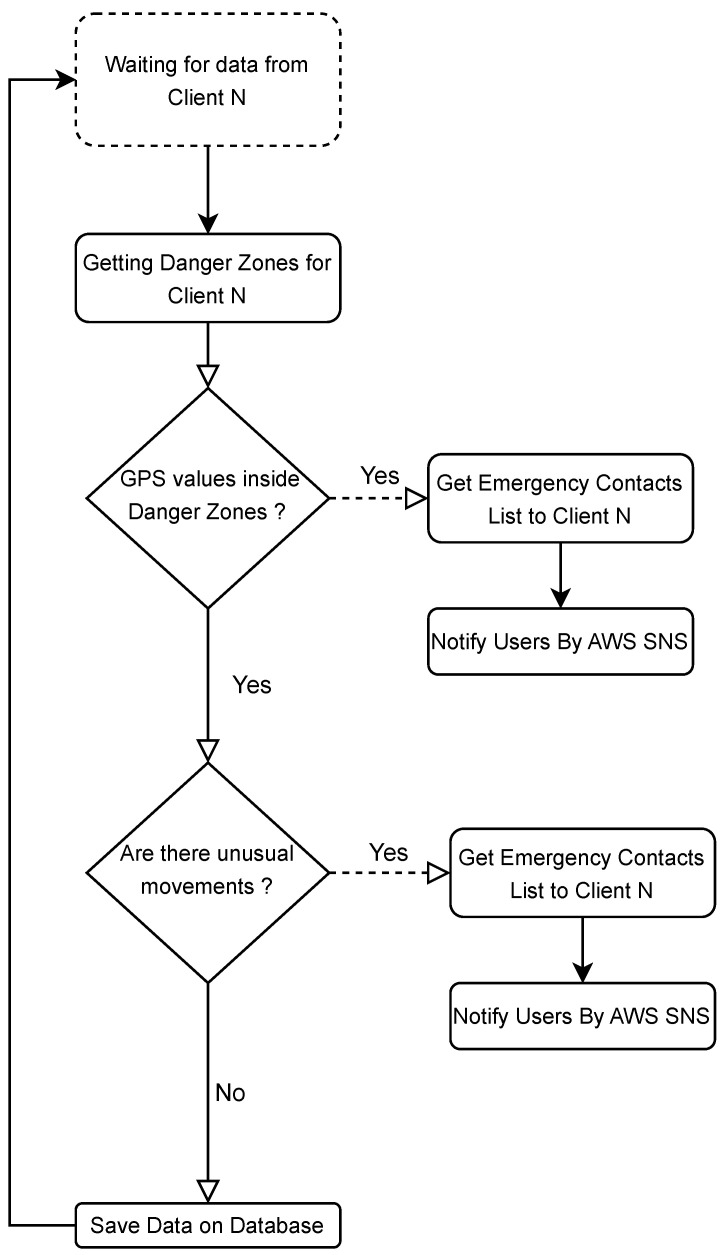
Flowchart of actions performed by the *Data Services*.

**Figure 8 sensors-23-07811-f008:**
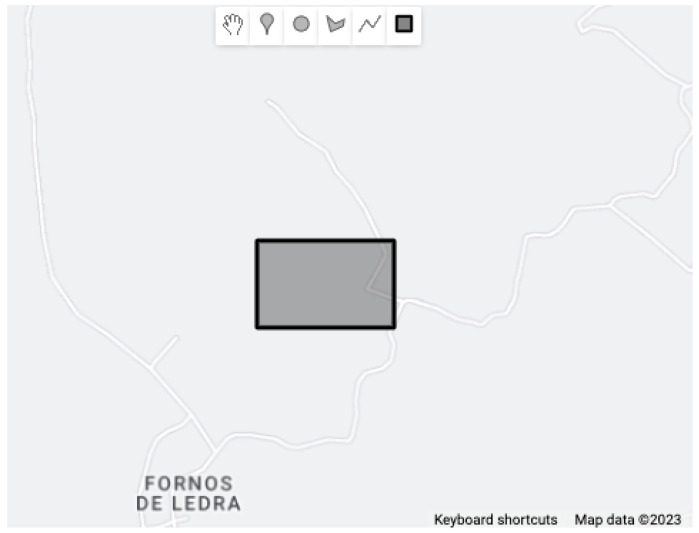
Definition of a danger zone.

**Figure 9 sensors-23-07811-f009:**
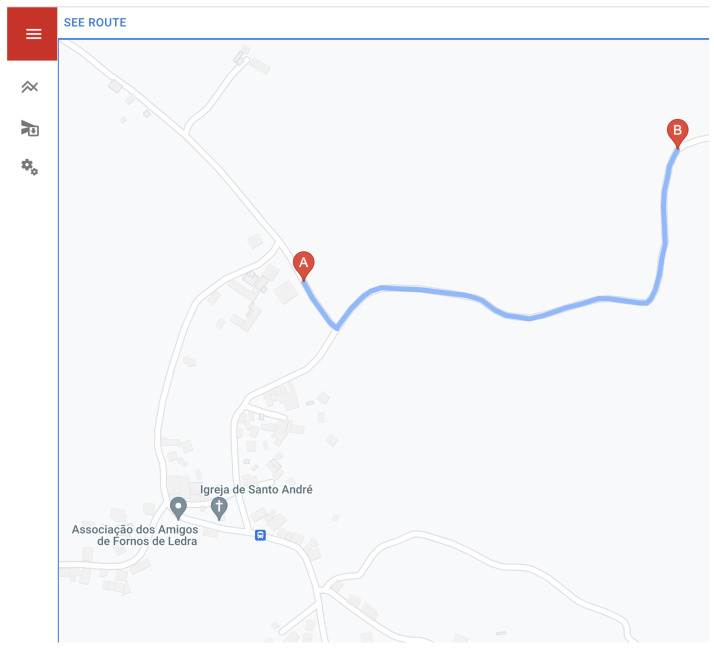
Reconstruction of the current farm tractor route.

**Figure 10 sensors-23-07811-f010:**
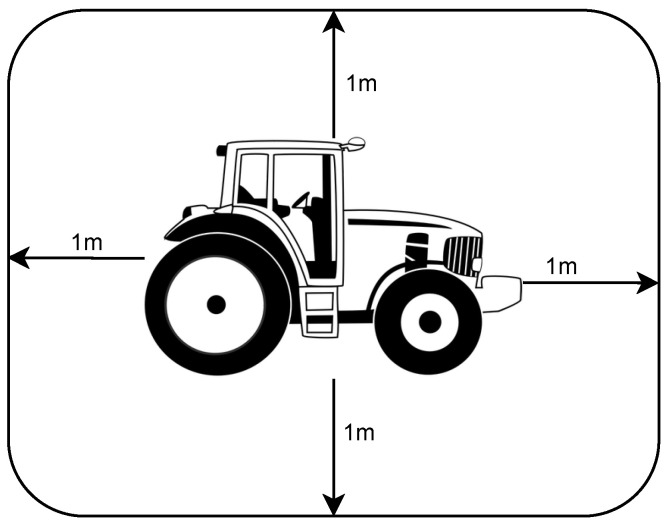
Protection zone of the tractor.

**Figure 11 sensors-23-07811-f011:**
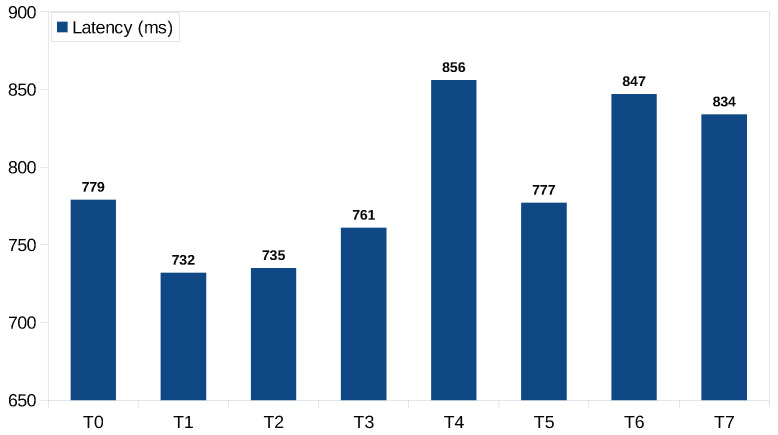
Latency results.

**Figure 12 sensors-23-07811-f012:**
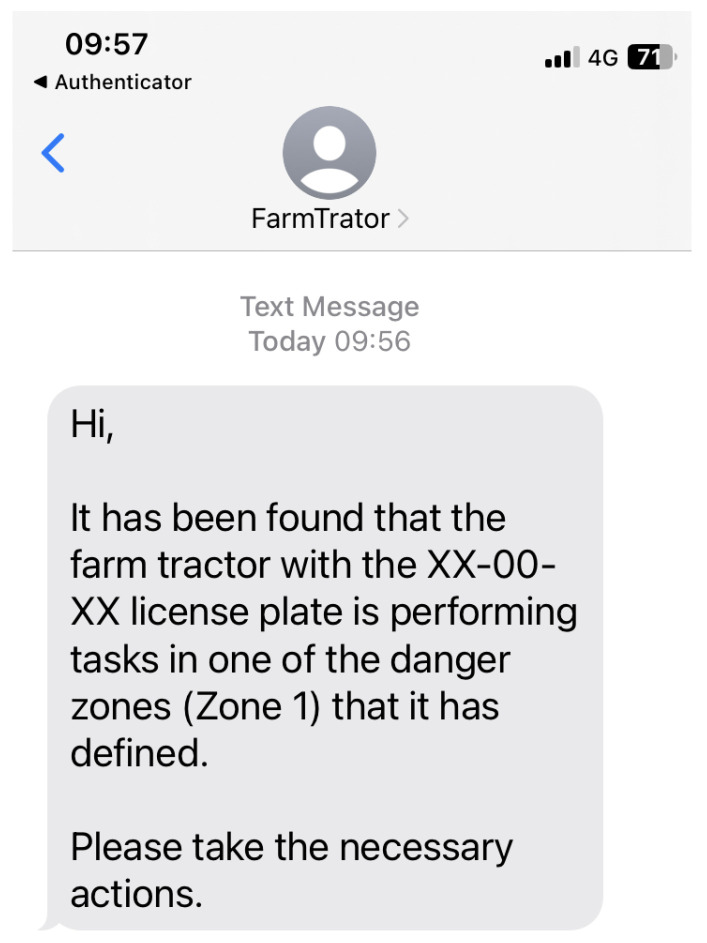
Example of an message to the emergency contact.

## Data Availability

Not applicable.

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
