# Peer review of "A Solution to Prevent and Minimize the Consequences of Accidents with Farm Tractors in the Context of Mountainous Regions with Low Population Density"

_sensors, 2023, doi:10.3390/s23187811_

Round 1
Reviewer 1 Report
Correlate it with other current Technologies, such as: Blockchain, IA, IoT (communications, networks, Cloud, …), in terms of latency I guess that this field is quite sensitive to the delays required to process data, which should call for new investigations around the tradeoff between learning cost and performance (e.g. Deep Learning is costly, yet attains good predictive scores… should we opt for weak learners over good features? Or complex learners over raw data? Or a mixture of both of them, e.g. learned features off-line + weak learners on-line? Should data be sent to the cloud? Be preprocessed at the edge?). This issue is also very trendy at the communications level.
Discussion should be placed along Opportunities and Open Issues in a single section (e.g., “Discussion, Opportunities and Open Issues”), and one open issue should be the lack of a thorough reference model for future developments, so that innovations in these platforms can be easily identified, exchanged, analyzed, exported, etc. In this way the contribution to the field would be very justified.
· The novelty of this paper is not clear. The difference between present work and previous Works should be highlighted.
· Experimental results are not clear. What are the parameters used in the proposed system and how their values are set? Also, how the parameter values can affect the proposed system? Sections like Experimentation have to be extended and improved thus providing a more convincing contribution to the paper.
· The authors provided details about the implementation setup and working environment. However, some training info should also be given in experimental section. How long does the proposed approach take to learn parameter? These details are missing and must be added to keep the paper standalone.
must be improved
Author Response
Please see the responses to the comments in the attached file - thank you.

Reviewer 2 Report
This paper is more like a technical report rather than a paper. I strongly suggest that the authors rewrite the contents and resubmit the paper. Some comments are listed as follows:
1. The topic of this paper should be revised, the authors are suggested to mention the main approach used.
2. The abstract must be rewritten as the authors devoted a large segment on the background of the farm tractor accidents. More description related to the approach should be discussed in the abstract.
3. The motivation and objective of this paper are not clear. What are the limitations of existing accident monitoring systems, and how the proposed approach can address these concerns? Besides, the contributions of this paper are not discussed in the Introduction section.
4. More related studies should be overviewed in the Related Work section. Besides, the authors are suggested to compare the proposed architecture with these existing systems.
5. Section 4 lacks the performance evaluation of the proposed approach. Comprehensive experiments should be added, for example, the RTT and latency.
6. The English editing can be improved by some native English writers.
Extensive editing of English language required
Author Response

(The authors gave the same response as above.)

Round 2
Reviewer 1 Report
accepted as is
Reviewer 2 Report
The authors have incorporated all my concerns. There are no further requirements from my side.